# Complexity of non-trivial sound speed in inflation

Lei-Hua Liu[1, *] and Ai-Chen Li[2, 3, †]

[1]*Department of Physics, College of Physics, Mechanical and Electrical Engineering, Jishou University, Jishou 416000, China*
[2]*Institut de Ciències del Cosmos, Universitat de Barcelona, Martí i Franquès 1, 08028 Barcelona, Spain*
[3]*Departament de Física Quàntica i Astrofísica, Facultat de Física,*
*Universitat de Barcelona, Martí i Franquès 1, 08028 Barcelona, Spain*

In this paper, we study the impact of non-trivial sound on the evolution of cosmological complexity in inflationary period. The vacuum state of curvature perturbation could be treated as squeezed states with two modes, characterized by the two most essential parameters: angle parameter $\phi_k$ and squeezing parameter $r_k$. Through *Schrödinger* equation, one can obtain the corresponding evolution equation of $\phi_k$ and $r_k$. Subsequently, the quantum circuit complexity between a squeezed vacuum state and squeezed states are evaluated in scalar curvature perturbation with a type of non-trivial sound speed. Our results reveal that the evolution of complexity at early times shows the rapid solution comparing with $c_S = 1$, in which we implement the resonant sound speed with various values of $\xi$. In these cases, it shows that the scrambling time will be lagged with non-vanishing $\xi$. Further, our methodology sheds a new way of distinguishing various cosmological models.

## INTRODUCTION

With the development of AdS/CFT [1], the understanding of the nature of spacetime has been made significant progress, in which a bulk gravitational theory is equivalent to a CFT at its corresponding boundary. This conjecture can be dubbed as a particular realization of the holographic principle, consequently one can consider that gravity is an emergent phenomenon not as a fundamental force in nature. For the further development of this methodology, another well-known conjecture was proposed that the spactime comes via quantum entanglement [2], especially from a realizion called ER = EPR [3]. In light of this logic, it motivates us for exploring the nature of spactime from the perspective of quantum entanglement [4–8]. Thus, holographic entanglement has become a key step for investigating the essence of spacetime. However, Ref. [9] found that a boundary QFT will reach a thermal equilibrium within a short time, the evolution of its corresponding wormhole will cost much longer time comparing with QFT. Therefore, it motives that there is another quantum quantity named by complexity for describing the evolution of wormhole [10, 11]. More precisely, this evolution can be depicted by holographic complexity. For calculating complexity in light of holographic principle, Susskind and his collaborators have proposed two conjectures: CV conjecture (complexity=volum) and CA conjecture (complexity=action), in which the first one is translated into the computation of maximum volume of the wormhole in bulk space and the second is referred as the computation of action evaluated on a bulk subregion called Wheeler-DeWitt patch. After that, extensive studies of such theories have produced many profound results, see for example [12–19].

Motivated by the AdS/CFT, a hotspot research direction in the field of high energy physics is to investigate the physics of circuit complexity from viewpoint of quantum field theory. Currently, there are mainly two methods for calculating the complexity: a). Nielsen et al. are pioneering to propose a geometrical method for computing the complexity in phase space of quantum gates [20–22]; b). Under the framework of information geometry, one can use the "Fubini-Study distance" to proceed [23]. According to these two methods, one can use the wave functions in position space to explicitly obtain the complexity [24–26]. One can also use the covariance matrix to calculate the complexity [27–32]. Although the implication of complexity in high energy physics is still elementary, there are some attempts are investigated, such as the definition of complexity in QFT [33]. Even the Hawking radiation can be treated by this kind of quantum circuit complexity [34, 35]. More striking, the spacetime in some sense can be interpreted as the quantum circuit complexity [36].

In light of Nielson's geometrical method [20], it can be implemented into the early Universe, namely for the inflationary epoch. The essence of curvature perturbation is quantum perturbation, however, we are not capable of distinguishing that it is quantum or classical from observations. The standard procedure to deal with the circuit complexity is utilizing squeezed state satisfied with the minimal uncertain relation, thus it naturally utilizes this squeezed state as a ground state of curvature perturbation. Ref. [37] investigates the relation between the bounce Universe and complexity, in which they found that the post bounce was most signigicant and also gave a rough estimation of scrambling time scale. Similar work has been studied in [38], in which more inflationary models are studied and they found that the complexity in the matter dominant period (MD) is growing fastest. Once taking effects of background expansion into account [40], it is possible to study the evolution of circuit complexity for curvature perturbation where they found that the inflationary period has the most simple linear relation of complexity evolution and its value will be enhanced in later Universe. From another aspect, the complexity could be dubbed as an integral part of web of diagnostics

for quantum chaos [41–46] leading to the essential information about the scrambling time and Lyapunov exponent etc [43], in which it will provide the new perspective for exploring the cosmology.

For the cosmological field, it also motives us to explore the quantum complexity of quantum cosmology, especially for the very early universe [40]. In [37, 40], the quantum circuit complexity has been calculated for the scalar curvature perturbations in simple inflation model with constant sound speed, i.e. $c_S^2 = 1$. Alternatively, it is valuable to evaluate the deviation from the trivial should speed, we will consider the effects from non-trivial sound speed into the quantum circuit complexity, especially for the different partial differential equations of parameters of squeezed state also comparing with Ref. [37, 40]. Thus, our purpose is to study the effects of varying sound speed on the evolution of quantum circuit complexity in background of inflationary de-Sitter spacetime. In particular, we are interested in a type of sound speed resonance (SSR) [47]. This mechanism is of significant importantce since it can be realized in various aspects for enhancing the power spectrum for curvature perturbation and primordial gravitational waves [48–50]. On the other hand, the very recent Refs. [51, 52] has investigated the quantum curcuit complexity in k-essence inflation and BPS states, it shows the very different evolution of curcuit complexity. In order to capture the various chaotic features of SSR, the investigations of its evolution of complexity will lead to distinguish various cosmological models.

This paper is organized as follows. In section 2, we will review the non-trivial sound speed $c_S^2$ from the perspective of resonant production of PBH. In section 3, the squeezed state of cosmological perturvations will be given, in which there exists two parameters for this squeezed states. In section 5, the complexity of these cosmological perturbation will be obtained under the framework of geometrical method via Nielson's work. In section 7, we give our main conclusions and discussions.

## THE EFFECTS OF NON-TRIVIAL SOUND SPEED

Before recalling the non-trivial sound speed, we will consider the Friedmann-Lemaitre-Robertson-Walker background metric considered as our working metric,

$$ds^2 = a(\eta)^2(-d\eta^2 + d\vec{x}^2), \qquad (1)$$

where $a(\eta)$ is scale factor in conformal time and $\vec{x}$ denotes the spatial vector. Under this background, we could define the perturbation of some scalar field in inflationary period $\phi(x_\mu) = \phi_0(\eta) + \delta\phi(x_\mu)$ with its corresponding metric defined by

$$ds^2 = a(\eta)^2\bigg(-(1+\psi(\eta,x))d\eta^2 + (1-\psi(\eta,x))dx^2\bigg), \quad (2)$$

where $\psi(\eta, x)$ is the perturbation of metric. Being armed these two metrics and the perturbations of some scalar field, the perturbed action can be denoted by the curvature perturbation

$$S = \frac{1}{2}\int dt d^3x a^3 \frac{\dot{\phi}^2}{H^2}\bigg[\dot{\mathcal{R}}^2 - \frac{1}{a^2}(\partial_i\mathcal{R})^2\bigg], \qquad (3)$$

where $H = \frac{\dot{a}}{a}$, $\mathcal{R} = \psi + \frac{H}{\phi_0}\delta\phi$. Action (3) can be transferred into canonical normalized scalar field in light of the Mukhanov variable $v = z\mathcal{R}$ where $z = a\sqrt{2\epsilon}$ with $\epsilon = -\frac{\dot{H}}{H^2} = 1 - \frac{\mathcal{H}'}{\mathcal{H}^2}$,

$$S = \frac{1}{2}\int d\eta d^3x\bigg[v'^2 - (\partial_i v)^2 + \frac{z'}{z}v^2 - 2\frac{z'}{z}v'v\bigg], \quad (4)$$

where prime implies that the derivative with respect to the conformal time $\eta$ even for $\mathcal{H}$. In action (4), it clearly indicates the perturbation of curvature is of trivial sound speed namely, $c_S = 1$. To evaluate the influence of non-trivial sound speed for the complexity, we will introduce a simple mechanism of production of PBH from a resonant sound speed. In this new mechanism, the key ingredient for producing the enhanced value of curvature perturbation is the non-trivial sound speed $c_S$. For a general non-trivial sound speed [53, 54], it is defined by a canonical variable $v = z\zeta$ with $z = \sqrt{2\epsilon}a/c_S$ where $\epsilon$ has the same definition as the previous discussions. For the canonical variable, its corresponding Fourier mode $v_k$ satisfied with the following equation of motion,

$$v_k'' + (c_S^2 k^2 - \frac{z''}{z})v_k = 0, \qquad (5)$$

This equation of motion is our starting point for constructing the model. If the sound speed $c_S$ is one, it will nicely recover eom of curvature perturbation for slow-roll single-field inflation. As for the deviation from one of $c_S$, it experiences the non-canonical kinetic term, in which the inflation embedded into string theory [55, 56]. Meanwhile, this mechanism can be realized in DBI inflation [57]. From the perspective of effective field theory, when integrating out the heavy modes, it could also lead to the non-trivial sound speed $c_S$ [58].

Since the amplification of curvature perturbation is highly relevant with this non-trivial $c_S$. In order to realize the formation of PBH, the ehanced power spectrum at some certain scales is necesseary dubbed as one kind of mechanism. To characterize this enhanced power spectrum, Ref. [47] has proposed the sound speed is of the following form,

$$c_S^2 = 1 - 2\xi[1 - \cos(k_*\tau)], \qquad (6)$$

where $\xi$ is the amplitude of oscillation and $k_*$ is the frequency of oscillation. The motivation for exploring this specific sound speed is that it could reveal rich phenomelogical implications, in which this mechanism can be realized in various theoretical models, such DBI inflation,

even for the multifield inflation, *e.t.c.* Due to its extensive implication, we dub it as an important example of non-trivial sound speed to explore the evolution of curcuit complexity. Combining eq. (6) and eq. (5) and keeping the first order of $\xi$ for $\frac{z''}{z}$ since the value of $\xi$ is tiny compared with one. Subsequently, changing variable as $x = k_* \tau$, then eq. (5) will become Mathieu equation,

$$\frac{d^2 v_k}{dx^2} + (A_k - 2q \cos 2x) v_k = 0 \tag{7}$$

where $A_k = k^2/k_*^2 + 2q - 4\xi$ and $q = 2\xi - (k^2/k_*^2)\xi$. There is a key feature of the Mathieu equation, since it exists an instability region of eq. (7) depicting its exponential growth, in which it can be read by

$$v_k \propto \exp(\xi k_* \tau/2). \tag{8}$$

This exponential growth could lead to the amplification of curvature perturbation. Consequently, it will realize the generation of PBH during inflation. Observing that eq. (7) belongs to the narrow resonance due to the small value of $\xi$. And the resonance only occurs as $k = nk_*$ with $n$ is an integer. Lastly, we will evaluate the deviation from the standard sound speed, particularly the information from $\xi$. In figure (1), it clearly indicates that the de-

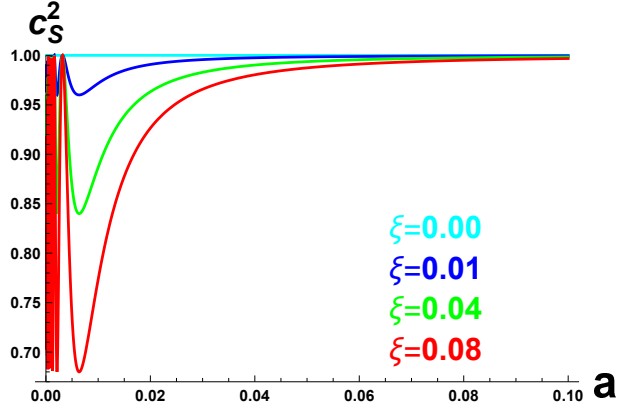

FIG. 1: It shows the resonant sound speed varying with scale factor $a(\tau)$ with various values of $\xi$. For a resonable input of $\xi$, we set its value are 0, 0.1, 0.04, 0.08, respectively.

viation from the standard sound speed ($c_S = 1$) varying with scale factor $a(\tau)$, in which it tells that there exists an oscillation at the very beginning of inflation ($a \approx 0.012$) and meanwhile this oscillation will be larger as enhancing the value of $\xi$, where we have used $\eta = -\frac{1}{Ha}$. From figure (1), we could also know that the parameter $\xi$ cannot be large due to the constraints of deviation from standard sound speed. Moreover, we will utilize this parameter to investigate its effects to the cosmological complexity.

## THE SQUEEZED QUANTUM STATES FOR COSMOLOGICAL PERTURBATIONS

In this paper, we will proceed with the complexity in inflationary period, namely for the quantum fluctuations. The complexity are usually depicting the chaotic and unstable feature of a statistical system. In the quantum level, the most simple and applicable system is the so-called inverted harmonic oscillator, in which its Hamiltonian is $H = \frac{1}{2}p^2 - \frac{1}{2}k^2x^2$ where $p$ is the momentum and $k$ is the frequency of oscillator. This inverted oscillator is an appropriate approximation for describing one chaotic and unstable system. Meanwhile, one could observe that the Hamiltonian operator of an cosmological system in momentum space is of the same mathematical structure as this inverted oscillator. This the reason we could use the methodology in quantum system into the cosmological system. The details of calculating the cosmological complexity are showing in Section 5. From another aspect, the wave function of the inverted quantum harmonic oscillator is the Gaussian wave function, thus the squeezed state is a natural way for realizing the evolution of the inflationary quantum curcuit.

For convenience, we will use the Mukhanov variable for depicting the action. To be more precise, EOM (5) is our starting point for calculating the complexity. To obtain EOM (5), its corresponding action can be explicitly written as

$$S = \int d\eta L = \frac{1}{2} \int d\eta d^3 x \left( v'^2 - c_S^2 (\partial_i v)^2 \right. \\ \left. + \left(\frac{z'}{z}\right)^2 v^2 - 2 \frac{z'}{z} v' v \right) \tag{9}$$

Here, the expression of $c_s^2$ is given by (6). The canonical momentum is obtained from (9)

$$\pi(\eta, \vec{x}) = \frac{\delta L}{\delta v'(\eta, \vec{x})} = v' - \frac{z'}{z} v \tag{10}$$

And then the Hamiltonian $H = \int d^3 x (\pi v' - \mathcal{L})$ is

$$H = \frac{1}{2} \int d^3 x \left[ \pi^2 + c_S^2 (\partial_i v)^2 + \frac{z'}{z} (v\pi + \pi v)) \right] \tag{11}$$

Promoting the Mukhanov variable $v$ to the quantum field and decomposing it into Fourier modes

$$\hat{v}(\eta, \vec{x}) = \int \frac{d^3 k}{(2\pi)^{3/2}} \frac{1}{\sqrt{2k}} \left( \hat{c}_{-\vec{k}}^\dagger v_k^\star(\eta) + \hat{c}_{\vec{k}} v_k(\eta) \right) e^{i\vec{k}\cdot\vec{x}} \tag{12}$$

Meanwhile, the corresponding $\hat{\pi}(\eta, \vec{x})$ is

$$\hat{\pi}(\eta, \vec{x}) = i \int \frac{d^3 k}{(2\pi)^{3/2}} \sqrt{\frac{k}{2}} \left( \hat{c}_{-\vec{k}}^\dagger u_k^\star(\eta) - \hat{c}_{\vec{k}} u_k(\eta) \right) e^{i\vec{k}\cdot\vec{x}} \tag{13}$$

where $\hat{c}_{-\vec{k}}^\dagger$ and $\hat{c}_{\vec{k}}$ represent the creation and annihilation operators respectively. By choosing an appropriate

normalization condition for mode functions $u_k(\eta)$, $v_k(\eta)$, one can give the following Hamiltonian

$$\hat{H} = \int d^3k \hat{\mathcal{H}}_k = \int d^3k \Big\{ \frac{k}{2}(c_S^2 + 1)\hat{c}^\dagger_{-\vec{k}}\hat{c}_{-\vec{k}}$$
$$+ \frac{k}{2}(c_S^2 + 1)\hat{c}_{\vec{k}}\hat{c}^\dagger_{\vec{k}} + \Big(\frac{k}{2}(c_S^2 - 1) + \frac{iz'}{z}\Big)\hat{c}^\dagger_{\vec{k}}\hat{c}^\dagger_{-\vec{k}}$$
$$+ \Big(\frac{k}{2}(c_S^2 - 1) - \frac{iz'}{z}\Big)\hat{c}_{\vec{k}}\hat{c}_{-\vec{k}} \Big\} \qquad (14)$$

In case of $c_S^2 = 1$, the (14) reduces to a Hamiltonian which is similar to the form of the inverted harmonic oscillator [39]. The unitary evolution operator acting on a state can be parameterized in the factorized form [59, 62]

$$\hat{\mathcal{U}}_{\vec{k}}(\eta, \eta_0) = \hat{S}_{\vec{k}}(r_k, \phi_k)\hat{\mathcal{R}}_{\vec{k}}(\theta_k) \qquad (15)$$

In (15), the $\hat{\mathcal{R}}_{\vec{k}}$ is the two-mode rotation operator written in term of the rotation angle $\theta_k(\eta)$

$$\hat{\mathcal{R}}_{\vec{k}}(\theta_k) = \exp\big[-i\theta_k(\eta)\big(\hat{c}_{\vec{k}}\hat{c}^\dagger_{\vec{k}} + \hat{c}^\dagger_{-\vec{k}}\hat{c}_{-\vec{k}}\big)\big] \qquad (16)$$

Meanwhile, $\hat{S}$ is the two-mode squeeze operator written in term of the squeezing parameter $r_k(\eta)$ and the squeezing angle $\phi_k(\eta)$ respectively.

$$\hat{S}_{\vec{k}}(r_k, \phi_k) = \exp\big[r_k(\eta)\big(e^{-2i\phi_k(\eta)}\hat{c}_{\vec{k}}\hat{c}_{-\vec{k}} - e^{2i\phi_k(\eta)}\hat{c}^\dagger_{-\vec{k}}\hat{c}^\dagger_{\vec{k}}\big)\big] \qquad (17)$$

Since the rotation operators only give an irrelevant phase factor when acting on the initial vacuum state, we will not include it in subsequent analysis. By acting the squeeze operator on the two-mode vacuum state $|0;0\rangle_{\vec{k},-\vec{k}}$, a two-mode squeezed state is obtained

$$|\Psi\rangle_{sq} = \frac{1}{\cosh r_k} \sum_{n=0}^{\infty} (-1)^n e^{2in\phi_k} \tanh^n r_k |n;n\rangle_{\vec{k},-\vec{k}} \qquad (18)$$

in which $|n;n\rangle_{\vec{k},-\vec{k}}$ represents the two-mode excited state

$$|n;n\rangle_{\vec{k},-\vec{k}} = \frac{1}{n!}\big(\hat{c}^\dagger_{\vec{k}}\big)^n\big(\hat{c}^\dagger_{-\vec{k}}\big)^n|0;0\rangle_{\vec{k},-\vec{k}} \qquad (19)$$

Together $(14),(18)$ with *Schrödinger* equation

$$i\frac{d}{d\eta}|\Psi\rangle_{sq} = \hat{H}|\Psi\rangle_{sq} \qquad (20)$$

We give the differeitial equations which control the time evolution of the squeezing paremeters $r_k(\eta)$, $\phi_k(\eta)$

$$-\frac{dr_k}{d\eta} = \frac{k}{2}(c_S^2 - 1)\sin(2\phi_k) + \frac{z'}{z}\cos(2\phi_k) \qquad (21)$$

$$\frac{d\phi_k}{d\eta} = \frac{k(c_S^2 + 1)}{2} - \frac{k}{2}(c_S^2 - 1)\cos 2\phi_k \coth 2r_k$$
$$+ \frac{z'}{z}\sin 2\phi_k \coth 2r_k \qquad (22)$$

These two equations are difficult to solve analytically, thus we have to consider the numerical solutions. For the sake of simplicity for calculation, the variable $\log_{10} a$ are utilized to take place of the conformal time $\eta$. Under this variable transformation, together with the expression (6) and $z = \frac{\sqrt{2}\epsilon a}{c_S}$, equations (22) are expanded as

$$\frac{10^y H_0}{\ln[10]} \cdot \frac{dr}{dy} = \Big(\frac{2k_\star \xi \sin(\frac{2k_\star}{10^y H_0})}{1 - 2\xi + 2\xi \cos(\frac{2k_\star}{10^y H_0})} - 10^y H_0\Big) \cdot \cos(2\phi_k)$$
$$+ k\xi\big(1 - \cos(\frac{2k_\star}{10^y H_0})\big) \cdot \sin(2\phi_k) \qquad (23)$$

$$\frac{10^y H_0}{\ln[10]} \cdot \frac{d\phi_k}{dy} = k\Big(1 - \xi\big(1 - \cos(\frac{2k_\star}{H_0 \cdot 10^y})\big)\Big)$$
$$+ \Big(10^y H_0 - \frac{2k_\star \xi \sin(\frac{2k_\star}{10^y H_0})}{1 - 2\xi + 2\xi \cos(\frac{2k_\star}{10^y H_0})}\Big) \cdot \sin 2\phi_k \coth 2r_k$$
$$+ k\xi\big(1 - \cos(\frac{2k_\star}{H_0 \cdot 10^y})\big) \cdot \cos 2\phi_k \coth 2r_k \qquad (24)$$

in which we have denoted $y = \log_{10} a$. Being armed with this variable transformation, one can obtain the numerical solution of $\phi_k$ and $r_k$ depicted by figure (2). Figure

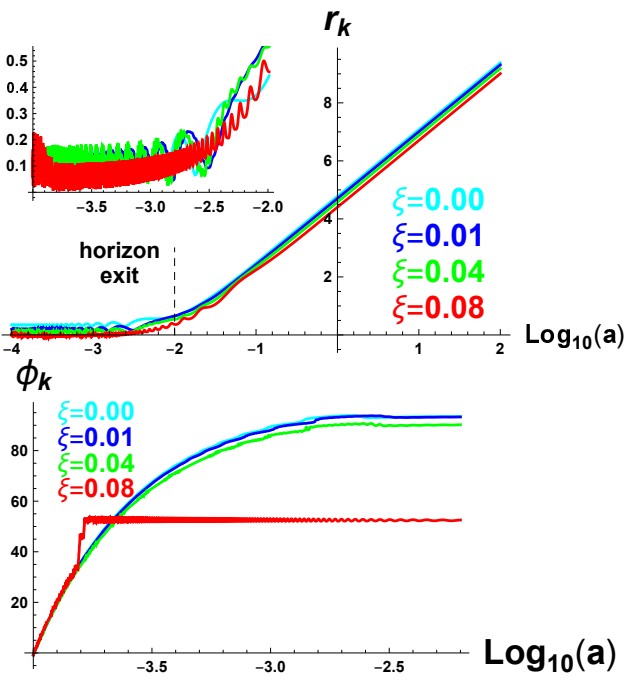

FIG. 2: The numerical solutions of $\phi_k$ and $r_k$ in terms of $\log_{10}(a)$ with $\xi = 0$, $\xi = 0.01$, $\xi = 0.04$ and $\xi = 0.08$. Our plots adopts $H_0 = 1$.

(2) shows the evolution of $\phi_k$ and $r_k$ with various values of parameter $\xi$. In the upper panel of figure (2), the trend of numerical solutions of $r_k$ with various $\xi$ is almost the same, particularly for the case of $\xi = 0$ which will be recovered the solution of Ref. [37]. $r_k$ plays an essential role

in determining the trend of complexity showing in figure (3). Combining with the analysis for figure (3), our result indicates that linear growth for $\xi$ will be lagged with non-vanishing $\xi$, precisely speaking that the linear growing part will appear after horizon exit comparing with the case of $\xi = 0$ in [43], in which the linear growing part of $\xi = 0$ occurs at the horizon exit. This is a huge difference for resulting in the evolution of complexity highly relevant with the property of a quantum chaotic system. The second difference comes via the oscillation behavior at the very early universe, it shows an enhanced oscillation with increasing the value of $\xi$. Finally, all of these cases will be approached a nearly the same slope leading to the same evolution after all these cases exit the horizon. Generically, we recall that the definition of resonant sound speed $c_S$, in which the frequency of oscillation for resonant sound speed will also be enlarged as improving the value of $\xi$ indicating in figure (1). In some sense, the fast oscillation of non-trivial sound speed leads to a more rapid oscillation of $r_k$. As for $\phi_k$, the only difference occurs that in the case of $\xi = 0.8$, $\phi_k$ will be faster approaching a constant comparing with other cases. For a careful reader, it will be easily found that the constant will be smaller as decreasing the value of $\xi$ until $\xi = 0.8$ whose evolution will show an obvious difference. Being armed with these two parameters, we will analyze the evolution of complexity.

## THE COMPLEXITY OF COSMOLOGICAL SQUEEZED STATES

In this paper, we will evaluate the circuit complexity by using Nielsen's method [20–22]. First, a reference state $|\psi^R\rangle$ is given at $\tau = 0$. And then, we suppose that a target state $|\psi^T\rangle$ could be obtained at $\tau = 1$ by acting a unitary operator on $|\psi^R\rangle$, namely

$$|\psi^T\rangle_{\tau=1} = U(\tau = 1)|\psi^R\rangle_{\tau=0} \tag{25}$$

As usual, $\tau$ parametrizes a path in the Hilbert space. Generally, the unitary operator is constructed from a path-ordered exponential of a Hamiltonian operator

$$U(\tau) = \overleftarrow{\mathcal{P}} \exp\left(-i \int_0^\tau dsH(s)\right) \tag{26}$$

where the $\overleftarrow{\mathcal{P}}$ indicates right-to-left path ordering. The Hamiltonian operator $H(s)$ can be expanded in terms of a basis of Hermitian operators $M_I$, which are the generators for elementary gates

$$H(s) = Y(s)^I M_I \tag{27}$$

The coefficients $Y(s)^I$ are identified as the control functions that determine which gate should be switched on or switched off at a given parameter. Meanwhile, the $Y(s)^I$ satisfy the *Schrödinger* equation

$$\frac{dU(s)}{ds} = -iY(s)^I M_I U(s) \tag{28}$$

Then a *cost functional* is defined as follows

$$\mathcal{C}(U) = \int_0^1 \mathcal{F}(\mathcal{U}, \dot{\mathcal{U}})d\tau \tag{29}$$

The complexity is obtained by minimizing the functional (29) and finding the shortest geodesic distance between the reference and target states. Here, we restrict our attentions on the *quadratic* cost functional

$$\mathcal{F}(U, Y) = \sqrt{\sum_I (Y^I)^2} \tag{30}$$

In this work, the target state is the two-mode squeezed vacuum state (18). After projecting $|\Psi\rangle$ into the position space, the following wavefunction is implied [60, 61]

$$\Psi_{sq}(q_{\vec{k}}, q_{-\vec{k}}) = \sum_{n=0}^{\infty} (-1)^n e^{2in\phi_k} \frac{\tanh^n r_k}{\cosh r_k} \langle q_{\vec{k}}; q_{-\vec{k}}|n; n\rangle_{\vec{k}, -\vec{k}}$$

$$= \frac{\exp[A(r_k, \phi_k) \cdot (q_{\vec{k}}^2 + q_{-\vec{k}}^2) - B(r_k, \phi_k) \cdot q_{\vec{k}} q_{-\vec{k}}]}{\cosh r_k \sqrt{\pi}\sqrt{1 - e^{-4i\phi_k}\tanh^2 r_k}} \tag{31}$$

in which the coefficients $A(r_k, \phi_k)$ and $B(r_k, \phi_k)$ are

$$A(r_k, \phi_k) = \frac{k}{2}\left(\frac{e^{-4i\phi_k}\tanh^2 r_k + 1}{e^{-4i\phi_k}\tanh^2 r_k - 1}\right) \tag{32}$$

$$B(r_k, \phi_k) = 2k\left(\frac{e^{-2i\phi_k}\tanh r_k}{e^{-4i\phi_k}\tanh^2 r_k - 1}\right) \tag{33}$$

By using a suitable rotation in vector space $(q_{\vec{k}}, q_{-\vec{k}})$, the exponent in (31) could be rewritten by a form of diagonal matrix

$$\Psi_{sq}(q_{\vec{k}}, q_{-\vec{k}}) = \frac{\exp[-\frac{1}{2}\tilde{M}^{ab}q_a q_b]}{\cosh r_k \sqrt{\pi}\sqrt{1 - e^{-4i\phi_k}\tanh^2 r_k}} \tag{34}$$

$$\tilde{M} = \begin{pmatrix} \Omega_{\vec{k}'} & 0 \\ 0 & \Omega_{-\vec{k}'} \end{pmatrix} = \begin{pmatrix} -2A + B & 0 \\ 0 & -2A - B \end{pmatrix}$$

Meanwhile, the reference state is the unsqueezed vacuum state,

$$\Psi_{00}(q_{\vec{k}}, q_{-\vec{k}}) = \langle q_{\vec{k}}; q_{-\vec{k}}|0; 0\rangle_{\vec{k}, -\vec{k}}$$

$$= \frac{\exp[-\frac{1}{2}(\omega_{\vec{k}} q_{\vec{k}}^2 + \omega_{-\vec{k}} q_{-\vec{k}}^2)]}{\pi^{1/2}}$$

$$= \frac{\exp[-\frac{1}{2}\tilde{m}^{ab}q_a q_b]}{\pi^{1/2}} \tag{35}$$

$$\tilde{m} = \begin{pmatrix} \omega_{\vec{k}} & 0 \\ 0 & \omega_{-\vec{k}} \end{pmatrix}$$

Notice that (35) is the Guassian state related to the target state corresponding to the squeezed state via Hamilton operator (26), in which this operator will not change structure of the wavefunctions. Meanwhile, the squeezed state is one kind of Gaussian state, thus the reference state should also be a Gaussian state. According to the definition (25), one can associate the target state (34) with the reference state (35) through a unitary transformation

$$\Psi_\tau(q_{\vec{k}}, q_{-\vec{k}}) = \tilde{U}(\tau)\Psi_{00}(q_{\vec{k}}, q_{-\vec{k}})\tilde{U}^\dagger(\tau) \tag{36}$$

$$\Psi_{\tau=0}(q_{\vec{k}}, q_{-\vec{k}}) = \Psi_{00}(q_{\vec{k}}, q_{-\vec{k}}) \tag{37}$$

$$\Psi_{\tau=1}(q_{\vec{k}}, q_{-\vec{k}}) = \Psi_{sq}(q_{\vec{k}}, q_{-\vec{k}}) \tag{38}$$

where $\tilde{U}(\tau)$ is a $GL(2, C)$ unitary matrix which give the shortest geodesic distance between the target state and the reference state in operator space. As considered by the work [24], $\tilde{U}(\tau)$ will take the form

$$\tilde{U}(\tau) = \exp[\sum_{k=1}^{2} Y^k(\tau)M_k^{diag}] \tag{39}$$

where the $\{M_k^{diag}\}$ represent the 2 diagonal generators of $GL(2, C)$

$$M_1^{diag} = \begin{pmatrix} 1 & 0 \\ 0 & 0 \end{pmatrix} \ , \ M_2^{diag} = \begin{pmatrix} 0 & 0 \\ 0 & 1 \end{pmatrix}$$

Note that the off-diagonal components are set to zero since they will increase the distance between two states. The complex variables $\{Y^I(\tau)\}$ are constructed as [63]

$$Y^I(\tau) = Y^I(\tau = 1) \cdot \tau + Y^I(\tau = 0) \tag{40}$$

From the boundary conditions (37) and (38), one obtains

$$\text{Im}(Y^{1,2})\big|_{\tau=0} = \text{Re}(Y^I)\big|_{\tau=0} = 0 \tag{41}$$

$$\text{Im}(Y^{1,2})\big|_{\tau=1} = \frac{1}{2}\ln\frac{|\Omega_{\vec{k},-\vec{k}}|}{\omega_{\vec{k},-\vec{k}}} \tag{42}$$

$$\text{Re}(Y^{1,2})\big|_{\tau=1} = \frac{1}{2}\arctan\frac{\text{Im}(\Omega_{\vec{k},-\vec{k}})}{\text{Re}(\Omega_{\vec{k},-\vec{k}})} \tag{43}$$

And then the complexity could be rewritten as the following line length

$$C(\tilde{U}) = \int_0^1 d\tau\sqrt{G_{IJ}\dot{Y}^I(\tau)(\dot{Y}^J(\tau))^\star} \tag{44}$$

in which $G_{ij}$ is an induced metric for the group manifold. As indicated by the work [24], there exists an arbitrariness in term of the choice of $G_{IJ}$. In the previous illustration, it clearly indicates that $\tilde{U}(\tau)$ is of $GL(2, C)$ group structure, its corresponding geometry in operator space is a flat geometry, namely, $G_{IJ} = \delta_{IJ}$ which means that various group structures will lead to the different topology for its induced metric. Here, we should emphasize

that this induced metric is not the metric for the spacetime but for the operator parameter minifold. Thus, the complexity of eq. (44) can be applied into various cosmological background when $\tilde{U}(\tau)$ has the $GL(2, C)$ group structure. Substitute (40) into (44), the expression for the complexity is calculated as

$$\mathcal{C}(k) = \frac{1}{2}\Bigg(\Big(\ln\frac{|\Omega_{\vec{k}}|}{\omega_{\vec{k}}}\Big)^2 + \Big(\arctan\frac{\text{Im}(\Omega_{\vec{k}})}{\text{Re}(\Omega_{\vec{k}})}\Big)^2$$
$$+ \Big(\ln\frac{|\Omega_{-\vec{k}}|}{\omega_{-\vec{k}}}\Big)^2 + \Big(\arctan\frac{\text{Im}(\Omega_{-\vec{k}})}{\text{Re}(\Omega_{-\vec{k}})}\Big)^2\Bigg)^{1/2} \tag{45}$$

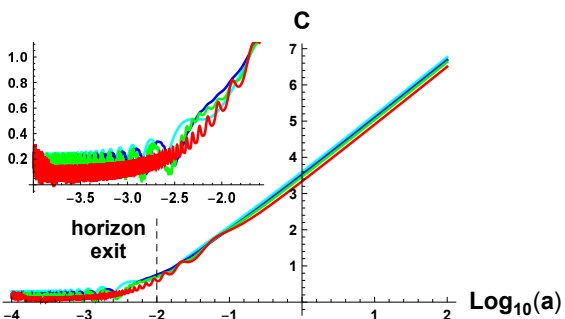

FIG. 3: Complexity is based on the numeical solution of $r_k$ and $\phi_k$. These colorful lines are corresponding to the previous figure (2). Our plots adopt $H_0 = 1$ and red curves for $\xi = 0.08$, green curve for $\xi = 0.04$, dark blue for $\xi = 0.01$, the light blue for $\xi = 0$ (corresponding to $c_S^2 = 1$). The larger value leads to the faster oscillation for complexity and will result in lagging the scrambling time. The scrambling time is considered as the complexity begins to increase and the Lyapunov index is identified with the growing of complexity becomes linear as mentioned in Ref [43].

From figure (3), the complexity is explicitly obtained via the numerical solution of $r_k$ and $\phi_k$, which indicates that the generic evolution of complexity. Combining with the analysis for figure (2), the huge difference comes via the linear growing part of the complexity. As [43] indicated, a quantum chaotic system is characterizing by the circuit complexity, especially for the time scale as the complexity begins to increase can be identified with scrambling time, also Ref. [43, 64] shows that the Lyapunov index can be approximately equaled to he slope of the linear growth. In light of their investigations and the above analysis showing in figure (2), the scrambling time will be lagged behind the time of horizon exit when increasing the value of $\xi$. Another difference is also for oscillation in the very early universe. Recalling that the formation of PBH in [47], it is generating at $k = nk_*$ in an inflationary period, at late times we cannot observe the formation of PBH. As a physical consequence, the corresponding evolution of complexity will not be changed agreed to our numerical simulation in figure (2). As for the inflationary period, we only consider the resonant

sound speed whose dynamical behavior will result in the oscillation of $c_S$ with the specific peak in the very early universe. Also as a physical consequence, the evolution of complexity will be oscillating faster in the very early universe. From the perspective of a chaotic system, the scrambling time (a time scale of a chaotic system) will be enhanced by adding the chaos in a chaotic system (the formation of PBH in our model) which is also agreed with our numerical simulation. Thus, it is reasonable to expect that our method could reveal a more fruitful evolution of complexity with various non-trivial sound speed inflationary models.

## SUMMARY AND DISCUSSION

The motivation to investigate the cosmological complexity is that it sheds a new way for distinguishing various inflationary models. As its practical realization, we evaluate the impact of non-trivial sound speed on the evolution of cosmological complexity in light of Nielson's geometrical method. Under the framework of standard cosmological perturbation theory and combining with the non-trivial sound speed, we utilize the squeezed state for investigations of curvature perturbation in the inflationary period. To achieve this goal, the most essential squeezed parameters are $r_k$ and $\phi_k$ in the phase space, respectively. Following the procedure, we obtain the evolution equation of $r_k$ and $\phi_k$ in terms of conformal time depicted by Eq. (22), in which there is a non-trivial sound speed term, in which we consider resonant sound speed as a special case. In a quantum chaotic system, the chaos is characterizing by the scrambling time and Lyapunov index. Meanwhile, Ref [43] has indicated that the time scale as the complexity begins to increase can be identified with scrambling time, particularly, the Lyapunov index is dubbed as the slope as the growing part of complexity becomes linear. In figure (3), our numerical simulation of complexity has clearly shown that the scrambling time will be lagged as enhancing the value of $\xi$, in which it was agreed with the reality since the formation of PBH will span a time scale of non-linear evolution of complexity. Another difference comes via the faster oscillation of complexity caused by this resonant sound speed.

Here, we emphasize the importance of investigations for the SSR belonging to the non-trivial sound speed. Firstly, the resonant sound speed (6) has extensive phenomelogical implications, as the previous discussions, this mechanism can be realized in DBI inflation, even for exploring the gravitatioal waves [48, 49] and analyzing the ehanced power spectrum of gravtational waves [50] under this mechanism. Since this mechanism can be realized in many models, the evolution of cosmological complexity for the resonant sound speed (6) will manifest the chaotic feature of these extensive models. In

[51, 52], it studied the evolution of cosmological complexity in k-essence inflationary model and BPS states, in which they show the various evolution for the cosmological complexity. Consequently, it sheds a new light for disguishing various cosmological models.

Next, we will utilize our method for investigating the more inflationary models with various non-trivial sound speed according to Eqs. 22, in which these two equations are valid for any non-trivial sound speed in inflation. In light of their solutions, we can simulate the evolution of complexity in different inflationary models. For analyzing the late time evolution of complexity, we need to take the potential into account. If translating into our method, there must exist extra terms in action (9) left to the future work. Furtherly, we could investigate the complexity from the modified gravity, especially for $f(R)$ gravity [65, 66] or the multifield inflation [67, 68], transformed into Einstein frame, there are non-trivial sound speed and possible extra terms of affecting the evolution of complexity. Finally, we could also work the complexity for a black hole Information loss problem solver [69] and the connection with Out-of-Time Ordered Correlators [70] (OTOC).

## Acknowledgements

LH is funded by NSFC grant NO. 12165009 and Hunan Provincial Department of Education, NO. 19B464. AC is supported by NSFC grant no.11875082 and the University of Barcelona (UB) / China Scholarship Council (CSC) joint scholarship. AC is also supported by National Natural Science Foundation of China under Grant Nos.11947086. We are grateful for the numerical calculation via Prof. Ding-Fang Zeng from Institute of Theoretical Physics In Beijing University of Technology

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
