# Peer review of "Complexity of non-trivial sound speed in inflation"

_SciPost Physics_

## Round 3 · Referee Report · Sayantan Choudhury · 2022-5-16

Strengths

All the computations and the results are presented in this paper are correct and presented in a very clearcut fashion. The related discussions of the obtained results are very precise and helpful to understand for general readers.

Weaknesses

The numerical part of the computation is not up to the mark. The authors need to focus on improving this part. I feel that a lot of improvement is required in this part of the computation in the paper. The related discussions are also not very clear. In short detailed discussion and precise numerical computations are required. Plots presented in this paper are not up to the mark and has to be improved considerably.

Report

1. The authors need to improve the numerical computations and plots. Also need to improve the related discussions and quality/presentation of the plots.
2. I also feel that the reference list is not complete. The authors need to cite more references on the related topic.
3. A short discussion on Krylov Complexity, which is a very new framework in this literature will be very helpful for understanding purpose. It is advised to include some discussions and its cosmological applications along with some references will suffice the purpose.

Requested changes

1. The authors need to improve the numerical computations and plots. Also need to improve the related discussions and quality/presentation of the plots.
2. I also feel that the reference list is not complete. The authors need to cite more references on the related topic.
3. A short discussion on Krylov Complexity, which is a very new framework in this literature will be very helpful for understanding purpose. It is advised to include some discussions and its cosmological applications along with some references will suffice the purpose.

---

## Round 3 · Author Response

Dear referee,
Thank you for your important report and sorry for the late response due to the family stuff for dealing with. As the report said, our paper lacks the theoretical motivation and importance of complexity for implementing into the cosmological perturbation theory. Due to this main point, I modify this my response for the report:
1 Complexity for quantum perturbations in the early Universe has been explored in previous literature. The main novelty of this work is to include a variable sound speed, satisfying a specific relation (equation 6). The authors conclude that the variable sound speed causes the scrambling time to lag, but this would surely be expected from the form of equation 6 (c_s^2 < 1 for non-zero xi). The authors should give motivations for the investigation undertaken in this work.
Our response: Combined with the second comment of this report, we emphasize that eq. (6) is not only one model for PBH formation, it can be realized in extensive models, thus the specific formula of (6) represent a large classes of phenomenological models. In order to depict the chaotic feature of this formula for the impact to the evolution of complexity, we also add the content in figure 3, in which it clearly shows that how the different values of xi affect the evolution, to be more precisely, it clearly indicates that how the different values of \xi will lag the scrambling time.
2.At a technical level, the authors do not explain for which models the speed of sound takes the form of equation 6 and why these models are of physical interest. In the conclusions the authors reference other inflationary models with different functional dependences for the sound speed and the focus on this specific model is not explained. The authors should explain why these models are important.
Our response: In introduction part and conclusion part, we add the discussion why we should investigate the non-trivial sound speed models, especially for the sound speed resonance (SSR) for which this mechanism of PBH formation can be realized in various models, thus if we study the evolution of this SSR mechanism, it could represent the chaotic feature of extensive models. Meanwhile, we also cite other references to illustrate the evolution of complexity in various models is different, thus it gives us the light for distinguishing the cosmological models. From another perspective, it also motivates us to explore the non-trivial sound speed comparing with c_S^2=1.
3.The main calculation of the complexity follows closely references [20-22, 24] but the applicability of these discussions to cosmological backgrounds is not discussed. The authors should explain why (35) is an appropriate reference state in this context, and explain the rationales for the implicit choices made in the cost functional (44). The authors should comment on whether different choices of the metric in (44) would give similar results.
Our response: The reference state should be of the Gaussian structure since the operator (26) upon acting on this reference state, it will not change the structure of wave function, and meanwhile the target state is squeezed state whose wave function is Gaussian state. In order to relate the reference state and target state via operator (26), we choose the reference state as eq. (35). As for the choice of metric (44), the induced flat metric is determined by the group structure of U(\tau) which is GL(2,C) leading to the flat metric and we should emphasize that the this metric is not the metric of spacetime. Consequently, we cannot freely choose the other induced metric in in which $G_{ij}$ is an induced metric for the group manifold. Here, we should emphasize that this induced metric is not the metric for the spacetime but for the operator parameter manifold.

---

## Round 3 · List of Changes

In the introduction part, in the fourth paragraph, we modify as following:

(1)Thus, our purpose is to study the effects of varying sound speed on the evolution of quantum circuit complexity in background of inflationary de-Sitter spacetime. In particular, we are interested in a type of sound speed resonance (SSR) [47]. This mechanism is of significant importance since it can be realized in various aspects for enhancing the power spectrum for curvature perturbation and primordial gravitational waves [48-50]. On the other hand, the very recent Refs. [51,52] has investigated the quantum circuit complexity in k-essence inflation and BPS states, it shows the very different evolution of circuit complexity. In order to capture the various chaotic features of SSR, the investigations of its evolution of complexity will lead to distinguish various cosmological models.

(2) In the conclusion part, we add one paragraph to emphasis the motivation of focusing on SSR (eq. (6)), it modified as following: “
Here, we emphasize the importance of investigations for the SSR belonging to the non-trivial sound speed. Firstly, the resonant sound speed (6) has extensive phenomenological implications, as the previous discussions, this mechanism can be realized in DBI inflation, even for exploring the gravitational waves [48,49] and analyzing the enhanced power spectrum of gravitational waves [50] under this mechanism. Since this mechanism can be realized in many models, the evolution of cosmological complexity for the resonant sound speed (6) will manifest the chaotic feature of these extensive models. In [51,52], it studied the evolution of cosmological complexity in k-essence inflationary model and BPS states, in which they show the various evolution for the cosmological models.

(3) After eq. (6), we also add the motivation for exploring this specific formula as following:” The motivation for exploring this specific sound speed is that it could reveal rich phenomenological implications, in which this mechanism can be realized in various theoretical models, such DBI inflation, even for the multi-field inflation”.

(4) In order to depic how non-vanishing \xi affect the lagging of srambling time, we give more illustration on the caption part of figure 3, we modify as following:“These colorful lines are corresponding to the previous figure \eqref{SolsqueezePara}. Our plots adopt $H_0=1$ and red curves for $\xi=0.08$, green curve for $\xi=0.04$, dark blue for $\xi=0.01$, the light blue for $\xi=0$ (corresponding to $c_S^2=1$). The larger value leads to the faster oscillation for complexity and will result in lagging the scrambling time. The scrambling time is considered as the complexity begins to increase and the Lyapunov index is identified with the growing of complexity becomes linear as mentioned in Ref [43]”

(5) For clarify the reference state (35) is gaussian state, we add the reason after eq. (35) as following: “Notice that (35) is the Gaussian state related to the target state corresponding to the squeezed state via Hamilton operator (26), in which this operator will not change structure of the wave-functions. Meanwhile, the squeezed state is one kind of Gaussian state, thus the reference state should also be a Gaussian state.”

(6) For why chossing flat metric of (44), we add the explanations as following:”In which $G_{ij}$ is an induced metric for the group manifold. As indicated by the work [24], there exists an arbitrariness in term of the choice of $G_{IJ}$. In the previous illustration, it clearly indicates that $\tilde{U}(\tau)$ is of $GL(2,C)$ group structure, its corresponding geometry in operator space is a flat geometry, namely, $G_{IJ}=\delta_{IJ}$ which means that various group structures will lead to the different topology for its induced metric. Here, we should emphasize that this induced metric is not the metric for the spacetime but for the operator parameter minifold. “

(7) For emphasize the importance of eq. (6), we add whole paragraph as the second paragraph in conclusion part, it is following ”Here, we emphasize the importance of investigations for the SSR belonging to the non-trivial sound speed. Firstly, the resonant sound speed (6) has extensive phenomenological implications, as the previous discussions, this mechanism can be realized in DBI inflation, even for exploring the gravitational waves [48,49] and analyzing the enhanced power spectrum of gravitational waves [50] under this mechanism. Since this mechanism can be realized in many models, the evolution of cosmological complexity for the resonant sound speed (6) will manifest the chaotic feature of these extensive models. In [51,52], it studied the evolution of cosmological complexity in k-essence inflationary model and BPS states, in which they show the various evolution for the cosmological complexity. Consequently, it sheds a new light for distinguishing various cosmological models.

You are currently on this page

Resubmission 2102.12014v3 on 28 January 2022

---

## Editorial Decision

editor-in-charge_assigned